# Infant Feeding Pattern Clusters Are Associated with Childhood Health Outcomes

**DOI:** 10.3390/nu15133065

**Published:** 2023-07-07

**Authors:** Ju Hee Kim, Eun Lee, Eun Kyo Ha, Gi Chun Lee, Jeewon Shin, Hey-Sung Baek, Sun-Hee Choi, Youn Ho Shin, Man Yong Han

**Affiliations:** 1Department of Pediatrics, Kyung Hee University Medical Center, Seoul 02447, Republic of Korea; 2004052@gmail.com; 2Department of Pediatrics, Chonnam National University Hospital, Chonnam National University Medical School, Gwangju 61469, Republic of Korea; unelee@daum.net; 3Department of Pediatrics, Hallym University Kangnam Sacred Heart Hospital, Seoul 07441, Republic of Korea; dmsry1@gmail.com; 4Department of Computer Science and Engineering, Konkuk University, Seoul 05029, Republic of Korea; transpaction0518@gmail.com; 5Department of Pediatrics, Bundang CHA Medical Center, CHA University School of Medicine, 59, Yatap-ro, Seongnam 13496, Republic of Korea; jshin8@gmail.com; 6Department of Pediatrics, Hallym University Kangdong Sacred Heart Hospital, Seoul 05355, Republic of Korea; paviola7@gmail.com; 7Department of Pediatrics, Kyung Hee University Hospital at Gangdong, Kyung Hee University School of Medicine, Seoul 05278, Republic of Korea; chsh0414@naver.com; 8Department of Pediatrics, Yeouido St. Mary’s Hospital, The Catholic University of Korea, Seoul 07345, Republic of Korea; epirubicin13@gmail.com

**Keywords:** breastfeeding, cluster, children, disease, hospitalization

## Abstract

(1) Background: Feeding behavior habits have a pattern with a certain tendency during infancy. We aimed to identify the associations between feeding patterns in infancy and the subsequent 10-year childhood disease burden. (2) Methods: Data from 236,372 infants were obtained from the national health insurance and screening program records in South Korea. Parent-administered questionnaires during infancy provided details on the feeding type and types/frequency of complementary food for analyzing feeding patterns. The outcomes were all-cause hospitalization and the development of 15 representative childhood diseases until the age of 10 years. Anthropometric measurements obtained at 6 years of age were analyzed. To estimate outcome risks while considering multiple risk factors, we employed a Cox proportional hazard and modified Poisson regression. (3) Results: Three clusters were identified: high prevalence of breastfeeding with regular exposure to a variety of solid foods (*n* = 116,372, cluster 1), similar prevalence of breastfeeding and formula feeding with less exposure to solid foods (*n* = 108,189, cluster 2), and similar prevalence of breastfeeding and formula feeding with the least exposure to solid foods in infancy (*n* = 11,811, cluster 3). Compared with cluster 1, children in clusters 2 and 3 had increased risks of all-cause hospitalization (hazard ratio (HR), (95% confidence interval (CI)), 1.04 (1.03–1.06) and 1.08 (1.05–1.11), respectively). Children in clusters 2 and 3 had an increased risk of upper respiratory infection, pneumonia, and gastroenteritis, as well as neurobehavioral diseases. Overweight/obesity at the age of 6 years was associated with clusters 2 and 3. (4) Conclusions: Feeding patterns in infancy were associated with an increased risk of childhood disease burden.

## 1. Introduction

Infant feeding patterns affect childhood diseases, growth, and adiposity [1,2,3,4,5]. Furthermore, the diet in infancy affects the child’s growth and the development of diseases during childhood [6]. Breastfeeding in early life is important for infant nutrition and can have diverse health effects in later life [7]. The timing and type of the introduction of solid food in early life also affect life-long health [2,8]. In addition, the variety of food introduced in early life affects the nutritional status and childhood health [9]. After birth, eating habits are not fixed at a time point but rather have a pattern with a certain tendency [10]. Previous studies on the effects of diet patterns at specific ages on disease burden and obesity have allowed for the control of possible confounding factors, such as income and educational status. Because the diet pattern in early life is intricately composed of breastfeeding vs. formula feeding, the timing of food introduction, and types of solid food, care should be taken when interpreting the results of these types of studies.

Cluster analysis is an unsupervised learning algorithm that classifies subgroups, which share similar patterns, on the basis of variables selected according to the intention of the researchers, without assumptions on the likely relationships within the data [11,12]. The results of cluster analysis can provide information on the associations and patterns within the data. Studies on the potential impact of infant feeding practices, composed of several factors including the types of infant formula and the timing and types of solid food, on later health in childhood are lacking, and the results are unclear, partially due to its complexity. The application of cluster analysis in infant feeding practice has enabled the classification of a few mutually exclusive clusters based on the feeding type in early life. The identification of the association between each cluster of infant feeding patterns and childhood disease might help improve the health outcomes of children. Therefore, we aimed to identify clusters of infant feeding practices and investigate the associations of each cluster with childhood health outcomes.

## 2. Methods

### 2.1. Study Design and Data Source

This study involved the analysis of data from the merged database of the National Health Insurance Service (NHIS) and the National Health Screening Program for Infants and Children (NHSPIC) in South Korea, which constitutes a national representative data source of a single insurance system that covers nearly the entire population of South Korea [13]. The NHIS contains information on the baseline demographics (age, sex, insurance premium, living region) and healthcare utilization (type of hospital visits: emergency room, outpatient clinic, or hospitalization; illness at the time of each hospital visit based on the International Classification of Diseases 10th revision [ICD-10] codes; and prescribed medications). Furthermore, all children who were eligible for health insurance completed seven surveys in the NHSPIC, which includes surveys on general health, anthropometric exams, and physical examination from ages 4 to 71 months [13]. De-identified individual data were used for research purposes only, and this research was conducted with ethical clearance under the current National Health Insurance Act. The protocol of this study was reviewed and approved by the Institutional Review Board of the Korea National Institute for Bioethics Policy (P01-201603-21-005).

### 2.2. Study Population

As shown in Figure 1, of the children born between 2008 and 2009 in South Korea, 308,393 who completed both the first and second rounds of the NHSPIC at 4–6 and 9–12 months of age, respectively, and for whom nutritional information was available were included in the present study. Children with any one of the following criteria were excluded: (1) children who died, (2) birth weight < 2.5 or >4 kg, (3) multiple births, (4) prematurity, (5) perinatal diseases, (6) any congenital malformations or chromosomal abnormalities, (6) admitted to an intensive care unit (ICU) for more than 4 days before completing 1 year of age, or (7) surgery under general anesthesia within the first 1 year of life. Finally, 236,372 children were included in the present study and followed up until 9 years of age [7,13].

### 2.3. Feeding Pattern in Infancy

Questionnaires used in the first and second rounds of the NHSPIC provided information on the feeding pattern in infancy, which includes the type of feeding at young infancy and the variety and frequency of complementary foods. The questionnaires were answered by the parents or legal guardians of the children at the scheduled regular check-ups (4–6 and 9–12 months of age). The first question was “What do you primarily feed your baby?” in the first round of the NHSPIC. The answers were (1) only breastmilk, (2) only formula milk, and (3) mixed feeding of breastmilk and formula milk. The second question was “Did you give your baby the following foods of grains, vegetables, fruits, eggs, fish, and meat?” in the second round of the NHSPIC, and the answers for each type of food were yes or no. The last question was “How many times per day did you give complementary food to your baby?”, and the answers were (1) zero, (2) one, (3) two, (4) three, and (5) four or more. 

### 2.4. Primary Outcome

The primary outcome was the all-cause hospitalization and all-cause ICU admission for childhood disease burden after the age of 3 years. 

### 2.5. Secondary Outcome: Childhood Diseases

We chose 15 representative pediatric diseases: upper airway disease (adenotonsillectomy, chronic otitis media, and hospitalization for croup or upper respiratory tract infection (URI)), lower airway diseases (hospitalization for pneumonia, lower respiratory tract infection (LRI), or asthma), gastrointestinal diseases (hospitalization for gastroenteritis and irritable bowel syndrome), neuropsychiatric diseases (attention-deficit hyperactivity disorder, febrile convulsion, and epilepsy), immunological diseases (Kawasaki disease and idiopathic thrombocytopenic purpura), and cancer (Appendix A) [13,14]. 

### 2.6. Additional Outcome: Overweight/Obesity at 6 Years of Age 

Weight and height were measured at 66–71 months of age at the seventh survey of the NHSPIC. Overweight and obesity were defined as the body mass index (BMI; calculated as weight (kg) divided by height (m) squared), based on the age z-score, ≥1.04 and ≥1.64, respectively [7].

### 2.7. Covariates

Information on sex (boy or girl), calendar year of the birth date, region of birth (Seoul, metropolitan, city, and rural), and economic status was obtained from the NHIS database, whereas data on the birth weight, body weight, and head circumference at 4–6 months were obtained from the NHSPIC database. Economic status was determined by the amount of the insurance co-payment, stratified into quintiles. In addition, we assessed perinatal conditions, including the fetal and neonatal impact of maternal conditions, birth trauma, respiratory and cardiovascular disorders specific to the perinatal period, infections specific to the perinatal period, fetal and neonatal hemorrhagic and hematological disorders, transitory fetal and neonatal endocrine and metabolic disorders, fetal and neonatal digestive disorders, conditions involving the fetal and neonatal integument and temperature regulation, and other disorders originating in the perinatal period. Moreover, comorbidities were defined based on hospitalization for wheezing, atopic dermatitis, and food allergy (Appendix A). 

### 2.8. Statistical Analysis

We conducted the Polytomous Variable Latent Class Analysis (poLCA) package (ver. 1.6.0.1) to determine the unique clusters of feeding patterns in infancy that were statistically independent with regard to a set of categorical variables [15,16,17]. Initially, we generated a series of models featuring a diverse range of latent clusters, spanning from two to ten. We subsequently evaluated the performance of each model, with the objective of determining the optimal fit for the data and the greatest possible distinction between the identified clusters. We utilized several statistical measures to evaluate the quality of the model fit, including the log likelihood plot, which indicates the point at which the log likelihood ceases to increase significantly, and the elbow heuristic for the Bayesian Information Criterion (BIC) and Akaike Information Criterion (AIC), where the change in successive values becomes less noticeable (Appendix A) [18,19,20,21]. To gauge the extent of the distinction between latent clusters, an entropy value was used, where a value of ≥0.6 was indicative of favorable separation between the groups [21,22]. Additionally, the minimum anticipated estimated class proportion was set to be no less than 5% to ensure meaningful findings [18,21,23]. In the final model, three clusters were identified as being best fitted.

All participants were followed up from the first year of age until the outcome of interest, death, or the end of the study (31 December 2018), whichever came first. Cox’s proportional hazard model was used to calculate the hazard ratio (HR) and the respective 95% confidence intervals (CI). The incidence rates were calculated as the sum of the expected events of interest per accumulated 1000 person-years (PY). The absolute rate differences with 95% CIs were calculated using a binomial regression model with a log-link function. To assess the association between feeding patterns in infancy and overweight/obesity at 6 years of age (as an additional outcome), a modified Poisson regression model was used to calculate the risk ratios (RRs) with 95% CIs. Furthermore, subgroup analyses of the HRs for all-cause hospitalization were calculated separately by sex (male vs. female), region at birth (Seoul/metropolitan vs. city/rural), socioeconomic status (low vs. high), birth year (2008 vs. 2009), and birthweight (≤3.2 kg vs. >3.2 kg). All analyses were adjusted by the variables of sex, region at birth, economic status, birthweight, body weight at 4–6 and 9–12 months of age, head circumference at 4–6 months of age, perinatal comorbidities, and comorbidities. 

All analyses were performed using the R package (ver. 4.1.3) and SAS version 9.4 (SAS Institute Inc., Cary, NC, USA). Two-sided *p*-values < 0.05 were considered statistically significant.

## 3. Results

### 3.1. Classification of Infant Feeding Clusters

Three clusters were identified in our study cohort: high prevalence of breastfeeding in early life with regular exposure to a variety solid foods in infancy (49.2%, *n* = 116,372, cluster 1), similar prevalence of breastfeeding and formula feeding in early life with less exposure to solid foods, especially protein-containing foods, in infancy (45.8%, *n* = 108,119, cluster 2), and similar prevalence of breastfeeding and formula feeding in early life with the least exposure to solid foods in infancy (5.0%, *n* = 11,811, cluster 3). Children in cluster 1 were introduced to 5.4 (standard deviation (SD) 0.3) types of solid food intake per day, whereas those in cluster 2 had 4.2 (SD 0.5) and those in cluster 3 had 3.8 (SD 0.9) food types. In addition, 81.0% of those in cluster 1 ate solid food three times a day, whereas only 57.5% and 42.8% of those in cluster 2 and cluster 3 did so. In infancy, breastfeeding at 6 months was noted in 52%, 43.4%, and 42.1% of clusters 1, 2, and 3, respectively (Table 1).

### 3.2. Characteristics of the Study Population

The characteristic features of each cluster are presented in Table 2. There was a difference in the sex ratio among the three clusters (*p* = 0.005); there was a higher proportion of girls than boys only in cluster 3. Compared to clusters 1 and 2, children in cluster 3 were more likely to be born in rural areas. In addition, socioeconomic status was relatively low in cluster 3 compared to that of clusters 1 and 2. Among the three clusters, the birthweight was the lowest in children in cluster 3, followed by those in cluster 2 and cluster 3. The body weight at 4–6 months of age and body weight and head circumferences at 9–12 months of age showed patterns similar to the birthweight. Comorbidities, including atopic dermatitis, food allergy, and hospitalization due to wheezing, were most prevalent in cluster 3, followed by cluster 2 and cluster 1. In addition, there were differences in the prevalence of perinatal comorbidities across the three clusters.

### 3.3. Association between All-Cause Hospitalization, ICU Care, and Feeding Patterns in Infancy

The associations of all-cause hospitalization and all-cause ICU admission with feeding patterns in infancy were investigated (Table 3). The incidence rate of all-cause hospitalization was 47.3, 50.5, and 54.9 per 1000 PY in clusters 1, 2, and 3, respectively. Compared to cluster 1, the risk of all-cause hospitalization was higher in cluster 2 (RD (95% CI), 3.16 (2.52–3.81)) and cluster 3 (RD (95% CI), 7.63 (6.05–9.21)). The hazard of all-cause hospitalization was 1.04 (95% CI, 1.03–1.06) in cluster 2 and 1.08 (95% CI, 1.05–1.11) in cluster 3. However, there was no difference in all-cause ICU admission across the three clusters (incidence rate per 1000 PY, 0.46, 0.46, and 0.52 in clusters 1, 2, and 3, respectively). Compared to that in cluster 1, the risk of all-cause ICU admission did not increase in cluster 2 (aHR (95% CI), 0.98 (0.86–1.10)) and cluster 3 (aHR (95% CI), 1.12 (0.85–1.47)). 

Furthermore, the risk of any-cause hospitalization was analyzed after stratification by sex, region at birth, economic status, birth year, and birth weight to confirm the consistency of the associations (Figure 2). In this subgroup analysis, all risks remained statistically significant and were consistent with the main findings.

### 3.4. Association between Specific Childhood Diseases and Feeding Patterns in Infancy

Figure 3 and Appendix A show the risk of 15 pre-specified childhood diseases in clusters 2 and 3 compared to cluster 1. The incidence rate of hospitalization due to URI and pneumonia per 1000 PY was 4.21 and 5.23 in cluster 3, 3.26 and 3.98 in cluster 2, and 3.03 and 3.70 in cluster 1, respectively. Thus, the risks of hospitalization due to URI and pneumonia were significantly increased in cluster 2 (aHR 1.06, 95% CI 1.01–1.11; aHR 1.07, 95% CI 1.03–1.12, respectively) and cluster 3 (aHR 1.31, 95% CI 1.19–1.44; aHR 1.35, 95% CI 1.24–1.47, respectively). However, there were no statistically significant differences in the associations of adenotonsillectomy, chronic otitis media, hospitalization due to croup, lower respiratory infection, or asthma among the three clusters.

In addition, the risk of hospitalization due to gastroenteritis increased in clusters 2 and 3 (aHR 1.04, 95% CI 1.01–1.07; aHR 1.17, 95% CI 1.09–1.24, respectively). However, there were no significant associations of irritable bowel syndrome with the infant feeding clusters.

With regard to neurologic and behavioral disorders, the risks of attention deficit hyperactivity disorder (ADHD), febrile convulsion, and epilepsy were significant in the intermediate- and poor-diet clusters. The risk of ADHD increased by 38% (95% CI, 22–57%) and 35% (95% CI, 19–53%) in cluster 3 and cluster 2. The HR of febrile convulsion was 1.07 (95% CI, 1.01–1.13) in cluster 2 and 1.22 (95% CI, 1.09–1.37) in cluster 3. In addition, the risk of epilepsy was significantly increased in cluster 2 (aHR, 1.18; 95% CI, 1.04–1.33) and cluster 3 (aHR, 1.43; 95% CI, 1.12–1.83). 

There were no significant differences in the incidence rates of immunologic disorders, including Kawasaki disorder and idiopathic thrombocytopenic purpura, or cancer across the three clusters.

### 3.5. Association of Overweight/Obesity at 6 Years of Age with the Infant Feeding Clusters

The associations of the infant feeding cluster with overweight/obesity at 6 years of age were identified (Figure 4 and Appendix A). Compared to children in cluster 1, children in cluster 2 and cluster 3 had a significantly increased risk of overweight and obesity. The aRRs for overweight in cluster 2 and cluster 3 were 1.07 (95% CI, 1.04–1.09) and 1.18 (95% CI, 1.12–1.24), respectively. In addition, the aRRs for obesity were significantly increased in cluster 2 (aRR, 1.09; 95% CI, 1.04–1.13) and cluster 3 (aRR, 1.27; 95% CI, 1.17–1.38).

## 4. Discussion

This study revealed consistent patterns in dietary behavior during the first year of life. Children in cluster 2 and cluster 3, who were characterized by a lower prevalence of breastfeeding in early life and a limited introduction of complementary foods during infancy, had a higher risk of all-cause hospitalization, childhood disease burden during a 9-year follow-up period, and obesity or overweight during preschool age. As this is the first study to investigate the association between infant diet patterns and diverse childhood disease outcomes, the findings provide valuable information for improving health outcomes in children. By highlighting the association between feeding patterns in infancy and long-term childhood disease burden, the findings of this study can help guide efforts to promote healthy feeding behavior during infancy and ultimately improve the overall health and well-being of children.

Breastfeeding has a protective role against childhood infectious diseases. Children who were breastfed exclusively for 15 weeks without the introduction of solid foods had a significant reduction in respiratory illness episodes during childhood [1]. Another study showed that exclusive breastfeeding for at least 3 months might have reduced the infectious disease-related morbidity of infants [5]. Exclusive breastfeeding in the first 6 months was associated with a decreased number of hospitalizations due to pneumonia [24]. In the present study, we identified that a similar prevalence of breastfeeding and formula feeding in early life with the lowest exposure to solid foods in infancy was associated with increased risks of hospitalization due to respiratory infectious diseases and gastroenteritis as well as the risk of any-cause hospitalization until 10 years of age. Although the results of the present study are similar to those of previous studies, our study is meaningful in that the results support the recommendation of breastfeeding in early life with regular exposure to a variety of solid foods in infancy for health outcomes during childhood.

Nutritional status during infancy plays a crucial role in brain development. In particular, undernutrition during infancy can affect cognitive development by causing direct structural damage to the brain and impairing infant motor system development and exploratory behavior [25]. However, studies of the potential impact of infant feeding on neurological diseases are scarce, and the findings of existing studies are not definitive [26,27,28]. A previous study reported a protective role of breastfeeding in febrile convulsion in the first year of life [27], and another study reported a small protective role of breastfeeding in febrile convulsions until 2 years of age [28]. However, no study has identified any associations of the diversity of infant diet with childhood neurologic diseases, including febrile convulsion, epilepsy, and ADHD. In our study, we found an association between the patterns of infant diet, including breastfeeding and complementary foods, and neurological diseases, such as febrile convulsion, ADHD, and epilepsy during childhood. Further research is needed to better understand the relationship between these factors and their underlying mechanisms.

We identified the high-risk infant dietary patterns of overweight and obesity in childhood, which were characterized by common formula feeding in the first 4–6 months and the inclusion of fewer types of complementary food and less frequent complementary food intake (cluster 3). It is well known that breastfeeding in infancy has a protective effect on obesity or overweight in childhood. In a meta-analysis of 17 studies, longer-duration breastfeeding was associated with a decreased risk of overweight or obesity in childhood [29]. In a randomized trial of more than 17,000 Belarusian infants, prolonged and exclusive breastfeeding was associated with a reduced risk of overweight and obesity at the age of 6.5 years [30]. In contrast, there are known effects of the timing of the introduction of complementary foods in infancy on the development of childhood overweight; however, the effects of introducing a varied and appropriate selection of complementary foods are not well understood. In children who are formula-fed and breastfed for less than 4 months, the introduction of complementary food before 4 months of age increased the risk of overweight in childhood [3]. In addition, the introduction timing of complementary food was associated with higher adiposity, which is different according to formula feeding and breastfeeding in early life [2]. Through our study’s results, it is clear that there is a need to confirm the relationship between the timing of the introduction of complementary food and the proper supplementation of complementary foods in infancy and overweight or obesity in adolescence.

To the best of our knowledge, this is the first study to comprehensively cluster the overall dietary habits of a large number of children during infancy and analyze the burden of childhood diseases over the following 9 years and the incidence of overweight at preschool age. To date, several studies have reported associations between infant feeding and childhood diseases and growth, including overweight and obesity, in children [1,2,3,5]. However, these studies had limitations in the identification of the effects of infant feeding on childhood health outcomes because they did not simultaneously consider complex infant feeding patterns, including the type and frequency of complementary foods as well as the introduction timing of complementary foods or the duration of breastfeeding. The simultaneous considerations of complex infant diet factors are indispensable for revealing the effect of an infant’s diet on the child’s subsequent health and growth. Therefore, we used a latent class analysis to determine the effects of diverse infant dietary patterns, which were intricately intertwined with the aspects of the duration of breastfeeding, the introduction timing of complementary foods, the frequency of complementary foods per day, and the types of complementary food, on childhood health and growth in the present study.

The timely introduction of nutritious solid foods plays a crucial role in promoting growth and development by providing sufficient nutrition [31]. Moreover, the intake of solid food affects the development of gut microbiota, contributing to the formation of a host-specific gut microbiome as it undergoes considerable changes from birth to 12–36 months of age under the influence of various factors, including dietary patterns [32,33]. The timing and nature of solid foods introduced early in life can cause alterations in the gut microbiome, subsequently shaping the immune system and thereby affecting childhood health, partially via immune-modulatory effects [33]. 

Nevertheless, the present study had several limitations. The scope of food items investigated in the present study was somewhat limited; for example, juices or teas were not included due to the use of a simplified questionnaire to foster a higher response rate from a large number of participants’ parents. Due to limitations in the availability of information, the results of the present study did not include the total calories and nutrients, which might have modified the effects of infant feeding patterns on health outcomes. In the present study, the measurement of food intake was not repeated during the follow-up period, which might have improved the accuracy of information on food intake. Information on childhood diseases was obtained using ICD-10 codes with prescription codes at the time of hospitalization to improve the accuracy of each disease outcome. To avoid confusion in the accuracy of disease diagnoses acquired using ICD-10 codes, we limited the health outcomes in the present study. Nevertheless, the results of the present study have important implications for improving childhood health outcomes.

In conclusion, infant feeding clusters based on breastfeeding in the first 4–6 months and the type and frequency of complementary food introduction and intake, respectively, are associated with an increased risk of childhood disease burden as well as overweight and obesity in preschool children. To improve childhood health outcomes, the varied and sufficient supplementation of complementary foods along with breastfeeding in the first 4–6 months are recommended. The results of the present study provide evidence of an association between infant feeding patterns and childhood disease burden.

## Figures and Tables

**Figure 1 nutrients-15-03065-f001:**
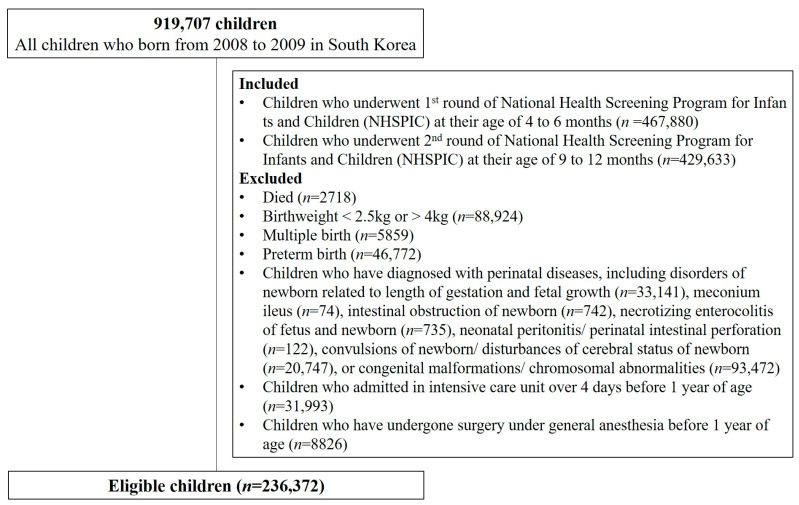
Study population.

**Figure 2 nutrients-15-03065-f002:**
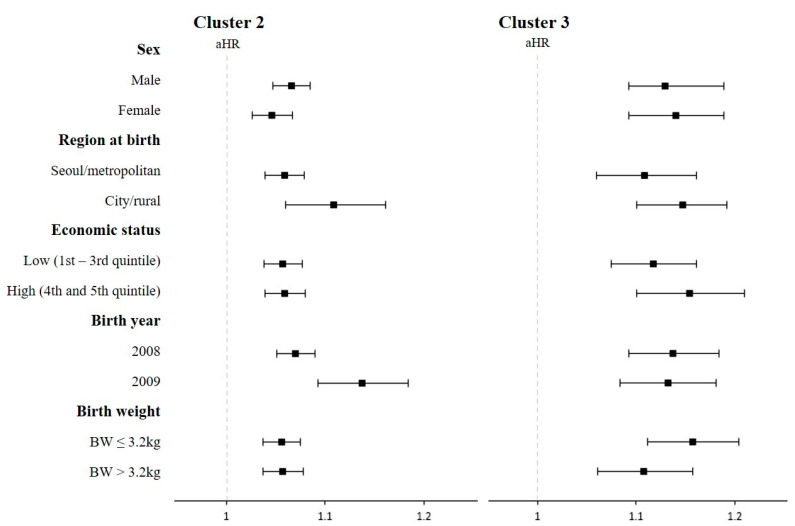
The risk of all-cause hospitalization based on the subgroup analysis stratified by baseline characteristics. Abbriviations: HR, hazard ratio; BW, birth weight. Children were classified based on feeding patterns in infancy by using polytomous variable latent class analysis. Adjusted hazard ratios and their 95% confidence intervals were calculated by a Cox proportional hazard model, with adjustment for sex, region at birth, economic status, birthweight, body weight at 4–6 and 9–12 months of age, head circumference at 4–6 months of age, perinatal comorbidities, and comorbidities. Filled squares indicate aHR and black lines indicate 95% CI.

**Figure 3 nutrients-15-03065-f003:**
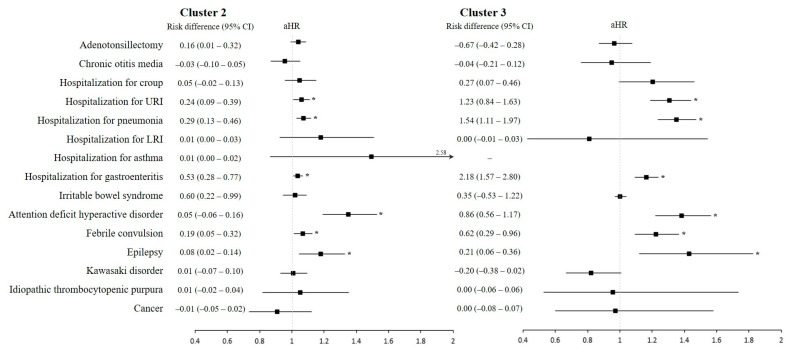
Risk of prespecified childhood diseases in the intermediate-diet and poor-diet clusters as compared to the healthy-diet cluster. Abbreviations: HR, hazard ratio; CI, confidence interval; URI, upper respiratory tract infection; LRI, lower respiratory tract infection. Children were classified based on feeding patterns in infancy by using polytomous variable latent class analysis. The definitions of each childhood disease were provided in Appendix A. Risk differences and their 95% confidence intervals were calculated by a binomial regression model with the log-link function. In addition, adjusted hazard ratios and their 95% confidence intervals were calculated by a Cox proportional hazard model, with adjustment for sex, region at birth, economic status, birthweight, body weight at 4–6 and 9–12 months of age, head circumference at 4–6 months of age, perinatal comorbidities, and comorbidities. Filled squares indicate aHR and black lines indicate 95% CI. Asterisks indicate *p* < 0.05.

**Figure 4 nutrients-15-03065-f004:**
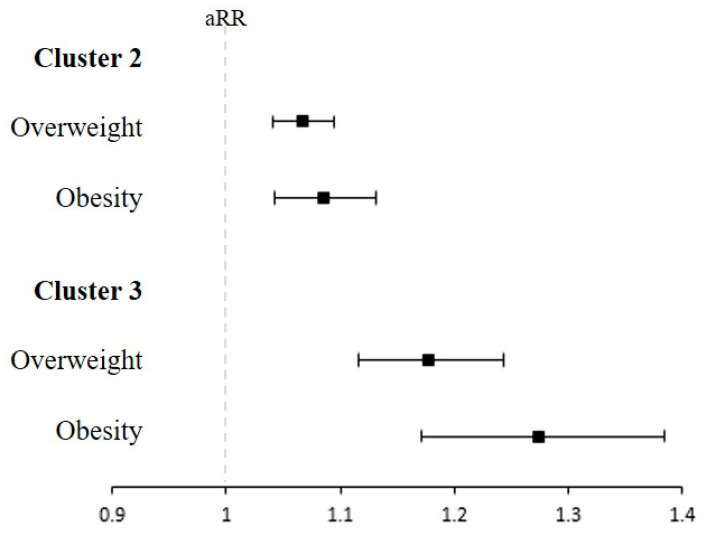
The risk of overweight/obesity at 6 years of age according to each infant feeding cluster. RR, Risk ratio. Children were classified based on feeding patterns in infancy by using polytomous variable latent class analysis. Overweight and obesity for age z-score were defined as body mass index (BMI) ≥ 1.04 and ≥1.64 at 6 years of age. Adjusted risk ratios and their 95% confidence intervals were calculated by using a modified Poisson regression model, with adjustments for sex, region at birth, economic status, birthweight, body weight at 4–6 and 9–12 months of age, head circumference at 4–6 months of age, perinatal comorbidities, and comorbidities. Filled squares indicate aHR and lines indicate 95% CI.

**Table 1 nutrients-15-03065-t001:** Feeding characteristics according to type of clusters ^1^.

Variables		Total	Cluster 1	Cluster 2	Cluster 3
Total number, *n* (%)		236,372 (100.0)	116,372 (49.2)	108,189 (45.8)	11,811 (5.0)
Types of solid foods introduced ^2^	Grains	209,265 (88.4)	113,467 (97.5)	91,525 (84.6)	4273 (36.2)
	Vegetables	223,621 (94.4)	115,588 (99.3)	105,421 (97.4)	2612 (22.1)
	Fruits	170,649 (72.1)	107,780 (92.6)	60,252 (55.7)	2608 (22.1)
	Eggs	130,266 (55.0)	105,257 (90.4)	23,331 (21.6)	1678 (14.2)
	Fish	129,197 (54.6)	99,778 (85.7)	28,270 (26.1)	1149 (9.7)
	Meat	213,764 (90.3)	114,486 (98.4)	98,547 (91.1)	731 (6.2)
Frequency of solid food intake per day ^2^	None	1967 (0.8)	247 (0.2)	327 (0.3)	1393 (11.8)
	1	5990 (2.5)	969 (0.8)	3407 (3.1)	1614 (13.7)
	2	60,030 (25.4)	16,427 (14.1)	40,195 (37.2)	3408 (28.9)
	3	161,471 (68.3)	94,228 (81.0)	62,193 (57.5)	5050 (42.8)
	≥4	6914 (2.9)	4501 (3.9)	2067 (1.9)	346 (2.9)
Types of feeding during the first 4 months of age ^3^	Only breastfeeding	112,277 (47.5)	60,376 (51.9)	46,931 (43.4)	4970 (42.1)
	Only formula milk feeding	76,992 (32.6)	31,850 (27.4)	40,319 (39.3)	4823 (40.8)
	Mixed feeding	46,146 (19.5)	23,678 (20.3)	20,549 (19.0)	1919 (16.2)

Abbreviations: *n*, number. ^1^ Children were classified based on feeding patterns in infancy using polytomous variable latent class analysis. ^2^ For children between 9 and 12 months, this questionnaire was answered by the parents or legal guardians in the second survey of the National Health Screening Program for Infants and Children. ^3^ For children between 4 and 6 months, this questionnaire was answered by the parents or legal guardians in the first survey of the National Health Screening Program for Infants and Children.

**Table 2 nutrients-15-03065-t002:** Baseline demographic and clinical characteristics according to cluster ^1^.

	Total (*n* = 236,372)	Cluster 1	Cluster 2	Cluster 3
Sex, *n* (%)				
Boy	120,066 (50.8)	59,131 (50.8)	55,086 (50.9)	5849 (49.5)
Girl	116,306 (48.2)	57,241 (49.2)	53,103 (49.1)	5962 (50.5)
Regions at birth, *n* (%)				
Seoul	58,815 (25.1)	30,234 (26.2)	26,422 (24.6)	2159 (18.5)
Metropolitan	55,217 (23.6)	26,854 (23.3)	25,716 (24.0)	2656 (22.7)
City	94,680 (40.4)	45,976 (39.9)	43,494 (40.6)	5210 (44.5)
Rural	25,559 (10.9)	12,299 (10.7)	11,590 (10.8)	1670 (14.3)
Socioeconomic status ^2^, *n* (%)				
First quintile (lowest)	17,863 (7.8)	8571 (7.6)	8172 (7.8)	1120 (9.9)
Second quintile	34,190 (15.0)	16,462 (14.6)	15,462 (14.8)	2266 (19.9)
Third quintile	63,557 (27.8)	30,819 (27.4)	29,348 (28.0)	3390 (29.8)
Fourth quintile	75,499 (33.1)	37,467 (33.3)	34,847 (33.3)	3185 (28.0)
Fifth quintile (highest)	37,271 (16.3)	19,073 (17.0)	16,800 (16.1)	1398 (12.3)
Birth weight ^3^, mean (SD), kg	3.2 (0.3)	3.2 (0.3)	3.2 (0.3)	3.2 (0.3)
Body weight at 4–6 months of age ^3^, mean (SD), kg	8.1 (1.0)	8.1 (1.0)	8.1 (1.0)	8.1 (1.0)
Body weight at 9–12 months of age ^4^, mean (SD), kg	9.8 (1.1)	9.9 (1.1)	9.8 (1.1)	9.8 (1.1)
Head circumference at 4–6 months of age ^3^, mean (SD), cm	42,750 (1.5)	42,768 (1.5)	42,741 (1.5)	42,656 (1.5)
Perinatal comorbidities ^5^, *n* (%)				
Birth trauma	2045 (0.9)	1049 (0.9)	908 (0.8)	88 (0.7)
Respiratory and cardiovascular disorders	9836 (4.2)	4778 (4.1)	4639 (4.3)	419 (3.5)
Infections	32,876 (13.9)	15,994 (13.7)	15,211 (14.1)	1671 (14.1)
Hemorrhagic and hematological disorders	74,439 (31.5)	36,632 (31.5)	34,246 (31.7)	3561 (30.1)
Transitory endocrine and metabolic disorders	5629 (2.4)	2830 (2.4)	2553 (2.4)	246 (2.1)
Digestive system disorders	6748 (2.9)	3186 (2.7)	3241 (3.0)	321 (2.7)
Integument and temperature regulation	8793 (3.7)	4200 (3.6)	4121 (3.8)	472 (4.0)
Comorbidities ^5^, *n* (%)				
Hospitalization due to wheezing	9476 (4.0)	4314 (3.7)	4493 (4.2)	669 (5.7)
Atopic dermatitis	36,093 (15.3)	16,058 (13.8)	17,991 (16.6)	2044 (17.3)
Food allergy	3721 (1.6)	1690 (1.5)	1819 (1.7)	212 (1.8)

Abbreviations: *n*, number; SD, standard deviation. ^1^ Children were classified based on feeding patterns in infancy by using polytomous variable latent class analysis. ^2^ Socioeconomic status was determined by the amount of the insurance co-payment, and the participants were stratified into quintiles. ^3^ Acquired or measured from the first survey of the National Health Screening Program for Infants and Children answered by their parents or legal guardians when the children were 4 to 6 months old. ^4^ Measured at the second survey of the National Health Screening Program for Infants and Children answered by their parents or legal guardians when the children were 9 to 12 months old. ^5^ Defined based on the ICD-10 codes in Appendix A. *p*-values for the differences between the three clusters were <0.05, except for birth trauma, hemorrhagic and hematological disorders, and transitory endocrine and metabolic disorders.

**Table 3 nutrients-15-03065-t003:** The risk of all-cause hospitalization and any-cause ICU admission stratified according to the cluster of feeding type during infancy ^1^.

	Cluster 1 (Reference) *n* = 116,372			Cluster 2 *n* = 108,189					Cluster 3 *n* = 11,811				
Diseases	*n* of Event	Accumulated *n*, 1000 PY	IR/1000 PY	*n* of Event	Accumulated *n*, 1000 PY	IR/1000 PY	RD ^2^ (95% CI)	aHR ^3^ (95% CI)	*n* of Event	Accumulated *n*, 1000 PY	IR/1000 PY	RD ^2^ (95% CI)	aHR ^3^ (95% CI)
All-cause hospitalization	44,572	942.6	47.29	43,578	963.8	50.45	3.16 (2.52 to 3.81)	1.042 (1.028 to 1.057)	5043	91.8	54.92	7.63 (6.05 to 9.21)	1.076 (1.046 to 1.109)
All-cause ICU admission	541	1169.7	0.46	500	1087.5	0.46	0.00 (−0.01 to 0.00)	0.975 (0.861 to 1.104)	62	118.7	0.52	0.06 (−0.16 to 0.15)	1.119 (0.853 to 1.469)

Abbreviations: *n*, number; PY, person-year; IR, incidence rate; RD, risk difference; aHR, adjusted hazard ratio; CI, confidence interval; ICU, intensive care unit; URI, upper respiratory infection; LRI, lower respiratory infection. ^1^ Children were classified based on feeding patterns in infancy by using polytomous variable latent class analysis. ^2^ Risk differences and their 95% confidence interval were calculated using a binomial regression model with the log-link function. ^3^ Adjusted hazard ratios and their 95% confidence intervals were calculated using a Cox proportional hazard model, with adjustments for sex, region at birth, economic status, birthweight, body weight at 4–6 and 9–12 months of age, head circumference at 4–6 months of age, perinatal comorbidities, and comorbidities. Bold indicates *p* < 0.05.

## Data Availability

This study was based on the National Health Claims Database established by the National Health Insurance Service of the Republic of Korea. Applications for using the National Health Insurance Service data are reviewed by the Inquiry Committee of Research Support; if the application is approved, raw data are provided to the applicant for a fee. We cannot provide access to the data, analytic methods, and research materials to other researchers because of the intellectual property rights of this database that is owned by the National Health Insurance Corporation. However, investigators who wish to reproduce our results or replicate the procedure can use the database, which is open for research purposes (https://nhiss.nhis.or.kr/ accessed on 7 July 2023).

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
