# Peer review of "Infant Feeding Pattern Clusters Are Associated with Childhood Health Outcomes"

_nutrients, 2023, doi:10.3390/nu15133065_

Round 1

Reviewer 1 Report

Very intriguing research question.  Much has been published on importance of breastfeeding (which these results support), and to a lesser degree the timing of complementary foods. But this investigation is the first I see that looks at the amount of complementary foods fed daily.  Study was soundly executed and results clearly presented.

While the authors addressed the study limitations, with respect to their statement "The number of investigated food items was limited in the present study...," I might suggest they insert, "For example, juices or teas were not included." 

Author Response

Reviewer 1. Very intriguing research question.  Much has been published on importance of breastfeeding (which these results support), and to a lesser degree the timing of complementary foods. But this investigation is the first I see that looks at the amount of complementary foods fed daily.  Study was soundly executed and results clearly presented.

Response: We extend our sincere gratitude to the editor and the reviewers for their meticulous evaluation of our manuscript. Additionally, we are appreciative of the positive feedback received regarding our research.

While the authors addressed the study limitations, with respect to their statement “The number of investigated food items was limited in the present study...,” I might suggest they insert, “For example, juices or teas were not included.” 

Response: We acknowledge and appreciate your constructive feedback. We have included the following statement (lines 126-129):

The scope of food items investigated in the present study was somewhat limited; for example, juices or teas were not included due to the use of a simplified questionnaire to foster a higher response rate from a large number of participants’ parents.

Reviewer 2 Report

This study analyzed the relationship between infant feeding practices and disease burden. It is the first to conduct a cluster analysis of the overall dietary habits of a large number of infants, providing the latest evidence that limited exposure to solid foods during infancy may increase disease burden. The study is innovative, with clear analytical methods and well-structured results, and it offers reasonable recommendations for improving child health outcomes through dietary improvements.

The current discussion primarily compares the findings with previous research results and elucidates the significance of this study. It would be helpful to further discuss the mechanisms by which introducing solid foods to infants significantly reduces the burden of certain diseases, as it could contribute to the improvement of the article.

Author Response

Reviewer 2. This study analyzed the relationship between infant feeding practices and disease burden. It is the first to conduct a cluster analysis of the overall dietary habits of a large number of infants, providing the latest evidence that limited exposure to solid foods during infancy may increase disease burden. The study is innovative, with clear analytical methods and well-structured results, and it offers reasonable recommendations for improving child health outcomes through dietary improvements.

Response: We extend our sincere gratitude to the editor and the reviewers for their meticulous evaluation of our manuscript. Additionally, we are appreciative of the positive feedback received regarding our research.

The current discussion primarily compares the findings with previous research results and elucidates the significance of this study. It would be helpful to further discuss the mechanisms by which introducing solid foods to infants significantly reduces the burden of certain diseases, as it could contribute to the improvement of the article.

Response: We appreciate your insightful comments. In light of your suggestions, we have incorporated a discussion on the potential mechanisms that underline the protective role of introducing healthy solid foods to infants and its subsequent impact on the disease burden during childhood as follows (lines 118-125):

Timely introduction of nutritious solid foods plays a crucial role in promoting growth and development by providing sufficient nutrition [37]. Moreover, the intake of solid food affects the development of gut microbiota, contributing to the formation of a host-specific gut biome as it undergoes considerable changes from birth to 12–36 months of age under the influence of various factors, including dietary patterns [38,39]. The timing and nature of solid foods introduced early in life can cause alterations in the gut microbiome, subsequently shaping the immune system and thereby affecting childhood health partially via immune-modulatory effects [39].

References

  1. Yu, C.; Binns, C.W.; Lee, A.H. The Early Introduction of Complementary (Solid) Foods: A Prospective Cohort Study of Infants in Chengdu, China. Nutrients 2019, 11, doi:10.3390/nu11040760.
  2. Homann, C.M.; Rossel, C.A.J.; Dizzell, S.; Bervoets, L.; Simioni, J.; Li, J.; Gunn, E.; Surette, M.G.; de Souza, R.J.; Mommers, M., et al. Infants' First Solid Foods: Impact on Gut Microbiota Development in Two Intercontinental Cohorts. Nutrients 2021, 13, doi:10.3390/nu13082639.
  3. Laue, H.E.; Coker, M.O.; Madan, J.C. The Developing Microbiome From Birth to 3 Years: The Gut-Brain Axis and Neurodevelopmental Outcomes. Front Pediatr 2022, 10, 815885, doi:10.3389/fped.2022.815885.
